# The structure and stability of $Fe_{4+x}S_3$ and its potential to form a Martian inner core

Lianjie Man [1] ✉, Xiang Li [2], Tiziana Boffa Ballaran[1], Wenju Zhou [3], Julien Chantel [4], Adrien Néri [1,4], Ilya Kupenko [2], Georgios Aprilis [2], Alexander Kurnosov [1], Olivier Namur[5], Michael Hanfland[2], Nicolas Guignot[6], Laura Henry [6], Leonid Dubrovinsky [1] & Daniel. J. Frost [1]

Seismic, geodetic and cosmochemical evidence point to Mars having a sulfur-rich liquid core. Due to the similarity between estimates of the core's sulfur content and the iron–iron sulfide eutectic composition at core conditions, it has been concluded that temperatures are too high for Mars to have an inner core. Recent low density estimates for the core, however, appear consistent with sulfur contents that are higher than the eutectic composition, leading to the possibility that an inner core could form from a high-pressure iron sulfide phase. Here we report the crystal structure of a phase with the formula $Fe_{4+x}S_3$, the iron content of which increases with temperature, approaching the stoichiometry $Fe_5S_3$ under Martian inner core conditions. We show that $Fe_{4+x}S_3$ has a higher density than the liquid Martian core and that a $Fe_{4+x}S_3$ inner core would crystalize if temperatures fall below 1960 (±105) K at the center of Mars.

Observations from NASA's InSight mission have revealed that the core of Mars is enriched in light elements, as its density appears to be substantially lower than that of Fe-Ni alloy[1–4]. Based on seismic wave reflections at the apparent core-mantle boundary of Mars, models considering either the existence[3,4] or absence[1–3] of a basal silicate magma layer indicate that the Martian core contains 9–20 wt.% or 20–25 wt.% of light elements, respectively. In either case, the abundance of light elements in the Martian core is significantly higher than in Earth's core (5–10 wt.%)[5], implying considerable differences in accretion and differentiation processes during the early stages of planetary formation[6]. From cosmochemical perspectives and geochemical considerations, candidate light elements in the Martian core include S, O, C, and H[7–10]. Sulfur, in particular, is often highlighted as a likely major light element in the Martian core, primarily due to it being the most prevalent moderately volatile element in the solar nebula[11], its siderophile ("iron-loving") behavior during core-mantle differentiation[12], and the fact that core formation on Mars was likely not a sufficiently reducing or high-temperature process for Si or O to be major light elements[13]. Assessments based on apparent depletions

of similarly volatile lithophile elements argue for <7 wt.% S in the Martian core[10] but this would most likely require significant proportions of C and H to explain the core's density deficit, which should, by the same arguments, be even more depleted in Mars than S. If depletions of similarly volatile elements are used to predict the S contents of ordinary and enstatite chondrites, the resulting concentrations for most of these meteorite sub-types are underestimated, raising the possibility that S contents of planetary bodies might vary independently of elements with similar condensation temperatures.

Seismic and lander radio science data from the InSight mission have confirmed that Mars has a liquid core[1–4,14], but the presence of a solid inner core cannot be currently excluded on geophysical grounds[1,2]. If further geophysical observations were to verify the existence, size, and density of a Martian inner core, then combined with the appropriate mineral physical interpretation, this would provide essential constraints on the composition and temperature of the interior, as well as the possible mechanisms that initiated and terminated the magnetic field of early Mars[15,16]. In the scenario of a S-rich Martian core, the cooling and solidification processes of an initially

¹Bayerisches Geoinstitut, Universität Bayreuth, Bayreuth, Germany. ²European Synchrotron Radiation Facility, Grenoble, France. ³Material Physics and Technology at Extreme Conditions, Laboratory of Crystallography, University of Bayreuth, Bayreuth, Germany. ⁴Univ. Lille, CNRS, INRAE, Centrale Lille, UMR 8207—UMET—Unité Matériaux et Transformations, Lille, France. ⁵Earth and Environmental Sciences, KU Leuven Leuven, Belgium. ⁶Synchrotron SOLEIL, L'Orme de Merisiers, Gif-sur-Yvette, France. ✉e-mail: lianjie.man@uni-bayreuth.de

fully molten Martian core are primarily governed by the melting phase relations of the Fe-FeS system under the high-pressure and high-temperature (HP-HT) conditions relevant to the Martian core. The eutectic composition in the Fe-FeS system shifts in the direction of the Fe-rich side with increasing pressure, from approximately 15.5 wt.% S at 21 GPa[17], i.e., the pressure at the top of the Martian core, to approximately 12 wt.% S at 40 GPa[18–21], the pressure at the center of Mars. Within the possible compositional range of Mars' core, either Fe or Fe sulfides could be liquidus phases that might crystalize as an inner core[16,18]. Understanding the crystal structures and densities of these liquidus phases is, therefore, critical for determining their behavior during cooling of the Martian core.

In addition to the endmembers Fe and FeS, the solid phases reported under Martian core conditions in the Fe-FeS system include $Fe_2S$, $Fe_3S$, and $Fe_{3+x}S_2$[17,18]. $Fe_2S$ is identified as a subsolidus phase in the Fe-FeS system at 21 GPa but is replaced by $Fe_3S$ or $Fe_{3+x}S_2$ when the temperature increases above the solidus temperature on the FeS-rich side of the eutectic[17]. Whether $Fe_2S$ becomes a liquidus phase under higher pressures, corresponding to deeper Martian core conditions, remains unknown. $Fe_3S$ adopts a $Fe_3P$-type structure and has a S content (16 wt.%) close to the eutectic composition of the Fe-FeS system at Martian core pressures[17]. During the cooling of the Martian core, if the sulfur concentration in the liquid core is above but close to the eutectic composition, $Fe_3S$ is expected to crystallize. $Fe_3S$ would be gravitationally stable at the center of the Martian core, as its density would be higher than that of the residual liquid[18]. However, if the bulk composition is more sulfur-enriched, for example, greater than 16 wt.% S at 21 GPa[17], the phase described as $Fe_{3+x}S_2$ would be the liquidus phase[16]. $Fe_{3+x}S_2$ decomposes during decompression and cannot be recovered to ambient pressure[22,23]. Its crystal structure can, therefore, only be investigated in situ, under high pressure conditions. The crystal system of $Fe_{3+x}S_2$ has been determined to be orthorhombic using powder X-ray diffraction[23], but its structure remains undetermined. Consequently, the density and elastic properties of $Fe_{3+x}S_2$ remain largely unknown.

In order to determine the crystal structure and density relations of the elusive $Fe_{3+x}S_2$ phase, we conducted a series of HP-HT experiments within the Fe-FeS system, employing multiple in situ and ex situ characterization techniques. However, instead of $Fe_{3+x}S_2$, we obtained a crystal structure for an iron sulfide phase that is more accurately described, on the basis of its crystallography, as $Fe_{4+x}S_3$. This phase was synthesized within the P-T and compositional range where $Fe_{3+x}S_2$ has been previously reported, which almost certainly has the same structure. We have further investigated the composition, density and potential role of $Fe_{4+x}S_3$ in forming a Martian inner core.

## Results

### Structural refinement of $Fe_{4+x}S_3$

As the target $Fe_{3+x}S_2$ phase is known to decompose to nano-crystallites of a few different phases during decompression[22], we performed high-pressure single crystal structure analyses in the diamond anvil cells (DAC) following in situ syntheses through laser heating (LH) at pressures of approximately 15 GPa and 21 GPa. The starting material for the LH-DAC experiments comprised degraded "$Fe_{3+x}S_2$" crystals, which were initially synthesized at pressures ranging from 14 to 16 GPa in a multi-anvil press. Although these starting materials maintained a homogeneous composition on a scale of less than 100 nm, their crystal structure experienced degradation during the decompression process in the MA press, and therefore, cannot be directly used for structure determination. After syntheses in the LH-DAC, single crystal X-ray diffraction (SC-XRD) data were collected at room temperature and high pressures on the newly grown crystals in the reacted LH area. Using a micro-focused synchrotron X-ray beam (-1 μm * 1 μm) it was possible to index reflections from numerous sub-μm sized grains with different orientations in each sample. Structural solution of several of

these grains led to the identification of a previously unknown structure exhibiting orthorhombic symmetry in the reacted areas of two experiments conducted at 15 GPa and 1150(±200) K (run LJFeS01) and 21 GPa and 1400( ± 200) K (run LD101). The reflections measured for all grains indicate clearly that the space group of this phase is *Pnma*. Structural refinement of two single crystals exhibiting the best discrepancy factors in each experiment (see more details in Supplementary Methods) indicates that the phase is characterized by five non-equivalent crystallographic sites for Fe and three non-equivalent sites for S.

As shown in Fig. 1, there are four Fe sites that are five-fold coordinated, forming four edge-sharing Fe-S square pyramids; whereas the remaining iron site is four-fold coordinated, creating a Fe-S tetrahedron. The fundamental building blocks of the structure of $Fe_{4+x}S_3$ are consistent with the high-pressure $Fe_{12}S_7$ phase[24], stable above 100 GPa, and the $Fe_2S$ phase stable above 21 GPa[24,25]. The tetrahedral site can be considered as an interstitial site between neighboring Fe-S square pyramids. If all the Fe and S sites are fully occupied, this would lead to a stoichiometry corresponding to $Fe_5S_3$. However, refinements of the SC-XRD data indicate that the Fe tetrahedral site is not fully occupied, leading to a chemical formula $Fe_{4+x}S_3$ (Supplementary Table 1 and Table 1), where x is the Fe occupancy at the tetrahedral site. In experimental run LD101, the occupancy of the tetrahedral site in the $Fe_{4+x}S_3$ phase was found to be 0.77 (±0.01), which can be described with the more appropriate stoichiometry $Fe_{4.77}S_3$, or following the formula of Fei et al.[17], $Fe_{3.18}S_2$. In another experimental run, LJFeS01, conducted under lower pressure-temperature (P-T) conditions than LD101, the tetrahedral site occupancy was refined to 0.11 (±0.01) Fe atoms. This results in the composition $Fe_{4.11}S_3$, or $Fe_{2.74}S_2$ when expressed in the $Fe_{3+x}S_2$ formula. Moreover, the unit cell volume of the $Fe_{4+x}S_3$ phase at 21 GPa and 300 K synthesized in run LD101 (331.2 Å³) is

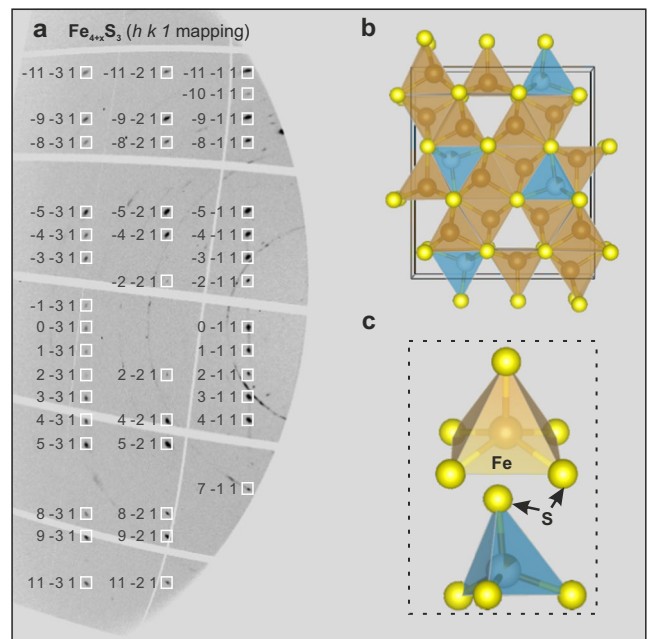

**Fig. 1 | In situ single crystal structure determination of $Fe_{4+x}S_3$ under high pressure. a** The diffraction mapping displays the reflections that satisfy the condition *h k* 1 for $Fe_{4+x}S_3$ (space group: *Pnma*), which were collected at 14.9(±1) GPa at room temperature following laser heating at 1150(±200) K (run LJFeS01). Miller indices are labeled alongside the reflections. The data were acquired through a step-scan procedure spanning the range of ω from −30° to 30°. **b** The structural model of $Fe_{4+x}S_3$ as determined by SC-XRD. **c** Depiction of the building blocks, including the Fe-S pyramid and semi-occupied Fe-S tetrahedron, that constitute $Fe_{4+x}S_3$. The crystal structure models were visualized using the software Vesta[67].

notably larger than that at 15 GPa and 300 K synthesized in run LJFeS01 (324.0 Å$^3$). This is clearly due to the increased Fe occupancy of the tetrahedral site, which causes an increase in tetrahedral volume and,

### Table 1 | Atomic coordinates and equivalent isotropic displacement parameters for Fe$_{4+x}$S$_3$

| | x | y | z | Occ. | Uiso |
|---|---|---|---|---|---|
| Fe$_{4.11}$S$_3$, 14.9(1) GPa, V = 324.0(6) Å$^3$ a = 10.897(5) Å, b = 3.125(1) Å, c = 9.515(18) Å | | | | | |
| Fe1 | 0.5700 (2) | 0.25 | 0.5822 (4) | 1 | 0.030 (2) |
| Fe2 | 0.2731 (3) | 0.25 | 0.0691 (5) | 1 | 0.026 (2) |
| Fe3 | 0.2847 (2) | 0.25 | 0.7910 (5) | 1 | 0.029 (2) |
| Fe4 | 0.0248 (2) | 0.25 | 0.6203 (5) | 1 | 0.036 (2) |
| Fe5 | 0.0632 (19) | 0.25 | 0.211 (4) | 0.11(1) | 0.033 (7) |
| S1 | 0.3726 (4) | 0.25 | 0.5862 (8) | 1 | 0.035 (3) |
| S2 | 0.3757 (4) | 0.25 | 0.2642 (8) | 1 | 0.036 (3) |
| S3 | 0.1277 (4) | 0.25 | 0.4200 (7) | 1 | 0.034 (2) |
| Fe$_{4.77}$S$_3$, 21.1(5) GPa, V = 332.4 (2) Å$^3$ a = 11.073(3) Å, b = 3.182(1) Å, c = 9.435(4) Å | | | | | |
| Fe1 | 0.5726 (2) | 0.25 | 0.5744 (3) | 1 | 0.017 (1) |
| Fe2 | 0.2781 (2) | 0.25 | 0.0618 (2) | 1 | 0.018 (1) |
| Fe3 | 0.2724 (2) | 0.25 | 0.7840 (3) | 1 | 0.019 (1) |
| Fe4 | 0.0283 (2) | 0.25 | 0.6084 (3) | 1 | 0.027 (1) |
| Fe5 | 0.0641 (2) | 0.25 | 0.2099 (3) | 0.77 (1) | 0.015 (1) |
| S1 | 0.3691 (3) | 0.25 | 0.5784 (4) | 1 | 0.015 (1) |
| S2 | 0.3743 (3) | 0.25 | 0.2629 (4) | 1 | 0.015 (1) |
| S3 | 0.1340 (3) | 0.25 | 0.4179 (4) | 1 | 0.018 (1) |

The space group of Fe$_{4+x}$S$_3$ is *Pnma*. The variables *x*, *y*, and *z* are the atomic coordinates, while *a*, *b*, c denote the unit cell parameters. *V* indicates the unit cell volume. "Occ." and "Uiso" indicate the site occupancy and isotropic displacement parameter, respectively. The numbers in parentheses are one standard deviation in terms of least units cited. All the data was collected at room temperature.

therefore, an increase in unit-cell volume (see Supplementary Discussion and Supplementary Fig. 1). The details of the crystallographic parameters are presented in Table 1 and Supplementary Table 2. The Fe$_{4.11}$S$_3$ crystal in run LJFeS01 was then further compressed up to 22.5 GPa at room temperature to examine its density and compressibility (Supplementary Table 3). The compression curve of Fe$_{4.11}$S$_3$ at room temperature was then fitted using a second-order Birch-Murnaghan equation of state (EOS)[26], resulting in $V_0 = 364.8(5)$ Å$^3$ and $K_0 = 97(2)$ GPa. After normalizing to the same pressure, for example, 21 GPa, the densities of Fe$_{4.11}$S$_3$ and Fe$_{4.77}$S$_3$ samples are 23.7% and 20.2% lower than hcp Fe[27], respectively. The non-stoichiometry, therefore, also affects the densiy, which increases with Fe content, in spite of the increase in unit cell volume.

To ascertain whether the Fe$_{4+x}$S$_3$ phase we identified at high pressure and room temperature is thermodynamically stable at the HP-HT conditions of synthesis, and to determine if a phase transition occurs during temperature quenching, we conducted in situ HP-HT XRD measurements using an Fe plus 15 wt.% S composition, in a multi-anvil press (MA) at the beamline PSICHE, SOLEIL (Supplementary Table 4). A representative result (run MA233), as presented in Fig. 2, reveals a series of peaks emerging in the energy-dispersive (ED) XRD pattern when the temperature reached 800 K at a pressure of approximately 14 GPa. These peaks, which cannot be indexed as polymorphs of Fe and FeS[28], can all be indexed to the *Pnma* Fe$_{4+x}$S$_3$ phase identified in our study. This finding supports the conclusion that the *Pnma* Fe$_{4+x}$S$_3$ phase is indeed the thermodynamically stable phase under these HP-HT conditions. The unit cell expanded by around 6% as the temperature increased from 800 K to 1100 K. This degree of expansion is too large to be solely attributed to thermal expansion. This abnormal volume expansion confirms the results of our LH-DAC experiments that the Fe$_{4+x}$S$_3$ phase tends to incorporate more iron and becomes progressively denser with increasing temperature.

The Fe$_{4+x}$S$_3$ phase discovered in this study is stable within the same P-T range reported for the Fe$_{3+x}$S$_2$ and Fe$_3$S$_2$ phases in the

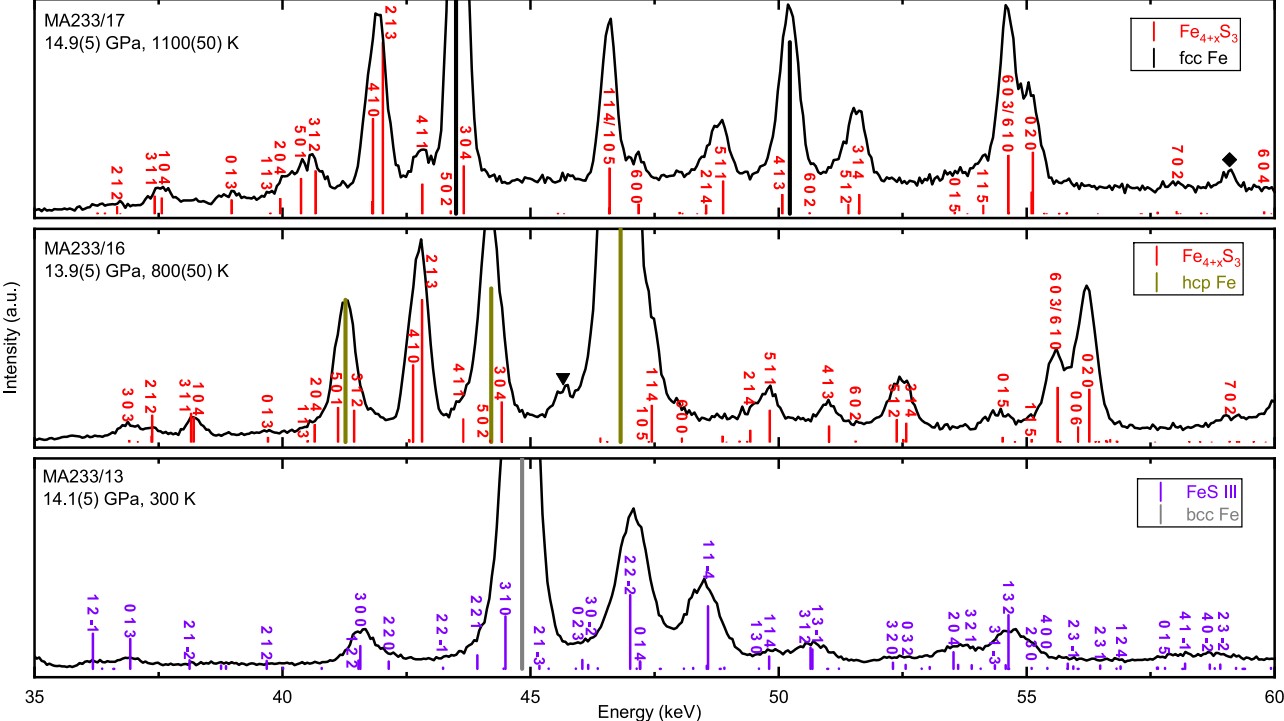

**Fig. 2 | In situ ED-XRD patterns collected in a synchrotron MA experiment using an Fe$_{85}$S$_{15}$ starting material conducted at approximately 14 GPa and at the temperatures indicated.** The Ge detector was positioned at an angle of 8.02 degrees 2θ. The grey, dark yellow, black, violet, and red lines indicate bcc Fe, hcp Fe, fcc Fe, FeS III, and Fe$_{4+x}$S$_3$ reflections, respectively. The minor peaks marked with black reversed triangles appear to be residual from FeS IV and those marked by diamonds are from FeO.

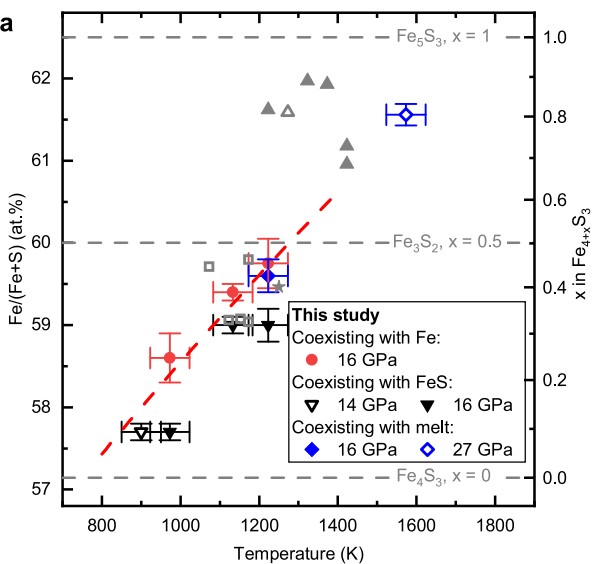
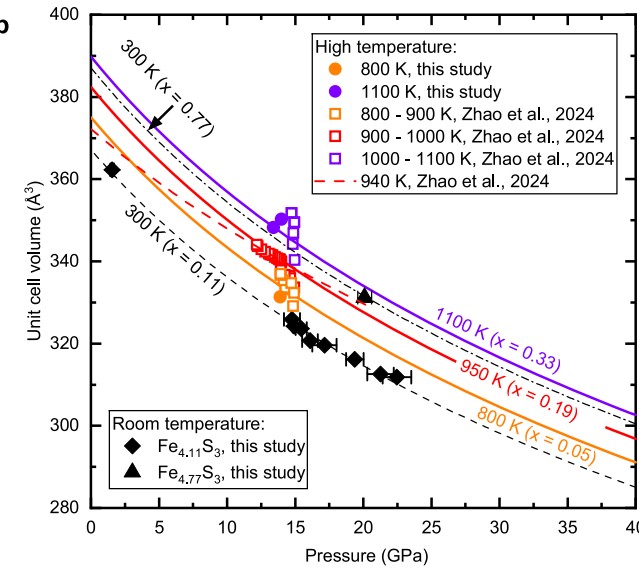

**Fig. 3 | P-V-T-x relations of Fe$_{4+x}$S$_3$. a** Compositions of Fe$_{4+x}$S$_3$ obtained from quenched MA experiments. In the present study, the Fe$_{4+x}$S$_3$ samples coexist with metallic Fe (red-filled circles, 16 GPa), solid FeS (black open inverted triangles, 14 GPa; black solid inverted triangles, 16 GPa), or Fe-S melt (blue solid diamonds, 16 GPa; blue open diamonds, 27 GPa). The grey symbols indicate results from the literature: open squares[30], filled triangles[17], open triangles[29], and the filled star[28]. The red dashed line is a linear fit of the x-T relationship for samples coexisting with metallic iron at pressures of 15–16 GPa. The error bars indicate the uncertainties in temperature measurements and the standard error (1σ) from the microprobe analyses. **b** Compression curves of Fe$_{4+x}$S$_3$ for various compositions and temperatures. The black dashed line,

orange solid line, red solid line, and violet solid line show values generated from the P-V-T-x model at 300 K, 800 K, 950 K, and 1100 K. The x values in high temperature curves follow the x-T relations indicated by the red dashed line in (**a**). The black diamonds and triangles are data collected in a DAC at 300 K in this study. The solid circles are the HP-HT data from this study and the open squares are the HP-HT data from the literature, which were previously interpreted as Fe$_3$S$_2$[23]. The red dashed line indicates the EOS of "Fe$_3$S$_2$" at 940 K, as reported by Zhao et al.[23]. The error bars indicate the uncertainties in pressure, stemming from the inaccuracies in volume determinations of the pressure markers. The uncertainties in the volume determinations of the samples are smaller than the size of the symbols shown.

literature[17,22,23,29–31]. The composition range and temperature-composition relations of Fe$_{4+x}$S$_3$ are consistent with that reported for both Fe$_{3+x}$S$_2$ and Fe$_3$S$_2$ (see Fig. 3a). Therefore, the Fe$_{3+x}$S$_2$ or Fe$_3$S$_2$ phase, whose crystal structure was previously unknown, is certain to be the same as the Fe$_{4+x}$S$_3$ phase identified in this study.

## P-V-T-x relations of Fe$_{4+x}$S$_3$

Chemical composition analyses of the quenched products from our in-house MA experiments further demonstrate the nonstoichiometric nature of the Fe$_{4+x}$S$_3$ phase, as well as its relationship with P, T, and iron activity. These experiments were carried out within the Fe-FeS system under a range of conditions: pressures from 14 to 27 GPa, temperatures from 918 to 1640 K, and varying bulk sulfur concentrations (Supplementary Table 5). Representative images illustrating the phase assemblages and textures of the recovered samples can be found in the Supplementary Fig. 2. Figure 3a illustrates that the Fe/(Fe+S) ratio, and consequently the value of x in Fe$_{4+x}$S$_3$, increases notably with increasing temperature. A large range of variation in the value of x in Fe$_{4+x}$S$_3$, from 0.09 to 0.80, was observed in this study that approaches the theoretical limits permissible within the crystallographic framework (i.e., 0 to 1). However, x also varies depending on the nature of the coexisting phase, i.e., the Fe activity, being approximately 0.2 higher when coexisting with metallic iron compared to FeS at 16 GPa, based on the results in this study. Although we cannot quantify the effect of pressure on the variation of x from the current dataset, x may increase with pressure at a given temperature. This possibility is implied by the deviation observed in the previous study by Fei et al.[17], conducted at 21 GPa, where x values are higher and deviate from the trend observed at 16 GPa in this study.

With the established relationship between temperature and composition for Fe$_{4+x}$S$_3$, we can evaluate its P-V-T-x relations using the HP-HT data from this study and from the literature[23]. The experiments from Zhao et al.[23] were conducted at pressures between 13 and 16 GPa,

and contain metallic iron. Therefore, we can fit the temperature-composition relationship from our in-house MA experiments and literature[28,30] with Fe$_{4+x}$S$_3$ coexisting with metallic iron at approximately 16 GPa to constrain the compositional effects (i.e., influence of the x parameter) on volume. A linear fit of $x = a \times (T - b)$ yields the result $a = 9.5 \pm 0.6 \times 10^{-4} \mathrm{K}^{-1}$ and $b = 750 \pm 50 \mathrm{K}$, where T is in K. The composition-volume relation can be determined using the single crystal refinements collected in this study, from the volume difference between the Fe$_{4.11}$S$_3$ and Fe$_{4.77}$S$_3$ samples, after normalization to the same pressure. After correcting the effects of composition and compressibility on the volume of Fe$_{4+x}$S$_3$, we can estimate its thermal expansion. We assume that the thermal expansion coefficient of Fe$_{4+x}$S$_3$ does not vary significantly with x, and fit the P-V-T data of Fe$_{4+x}$S$_3$ from this study and the literature[23] using the thermal expansion expression: $\alpha = 1/V(\partial V/\partial T)_P$, assuming $\alpha$ remains constant over the limited pressure (13–16 GPa) and temperature range (800–1100 K). The resulting thermal expansion coefficient is $5.3 \pm 2.0 \times 10^{-5} \mathrm{K}^{-1}$, which is comparable to that of the Fe$_3$S phase (-3.6 × 10$^{-5}$ K$^{-1}$ at 1000 K and 15 GPa[32]), but significantly smaller than that previously estimated for the "Fe$_3$S$_2$" phase (-26.58 × 10$^{-5}$ K$^{-1}$) in the same pressure and temperature range[23]. The significant overestimation of $\alpha$ in the previous study by Zhao et al.[23] is due to the fact that the volume expansion resulting from compositional variation with temperature was not accounted for separately in the evaluation of thermal expansion. The P-V-T-x relations of Fe$_{4+x}$S$_3$ are shown in Fig. 3. These relations accurately describe the large volume changes observed in the experimental data, which are caused by both thermal expansion and compositional variation.

## Discussion

Although there is currently no direct geophysical evidence confirming the existence of a Martian inner core, recent seismic and geodetic observations have provided important constraints on the state of the core as a whole[1–4,14]. Seismic measurements have detected the

apparent core-mantle boundary of Mars, supplied decisive evidence that at least the upper region of Mars' core is in a liquid state, and provided estimates for the core's average density[1–4] that range from 5.7 to 6.65 g/cm³. The substantial variation in these density estimates stems from whether a basal magma layer is considered to exit, which in turn implies different thermal regimes for Mars' interior. While the innermost state of Mars' core has not yet been revealed by seismic observations, the detected liquid region of the core sets an upper limit to a potential inner core radius of <750 km[2]. Models based on geodesy also support the existence of a liquid core, though these observations are generally insensitive to an inner core unless it is sufficiently large[33]. Based on the assumption that the sulfur content of Mars' core may be quite close to the Fe-FeS eutectic, previous models have proposed that temperatures are likely too high for an inner core to form[18,34]. However, the relatively low densities recently proposed for the core[1–3] raise the possibility that the composition may lie to the S-rich side of the eutectic at conditions approaching the center of Mars. To examine this

possibility, we first determine whether an $Fe_{4+x}S_3$ inner core would be gravitationally stable and the temperature required for it to crystallize, then compare this with proposed Martian areotherms to assess the likely core crystallization regime.

If we extrapolate the obtained P-V-T-x relations for $Fe_{4+x}S_3$ to inner core conditions, the density will increase both due to compression and because x will approach 1, reaching a value of 7.5(±0.3) g/cm³ at the center of Mars (40 GPa and 2000 K). This is larger than the range estimated using recent seismic observations for the density at the center of a liquid Martian core[1–4] (Supplementary Fig. 3), implying that an $Fe_{4+x}S_3$ inner core would be gravitationally stable in a bottom-up crystallization regime. It is worth noting that in estimating the density of $Fe_{4+x}S_3$, the thermal expansion coefficient was assumed to be constant, which likely results in a slight underestimate of the inner core density. Even though this assumption does not affect the conclusion of gravitational stability, further in situ HP-HT experiments would be required to establish a full thermodynamic model describing the P-V-T-x relations of $Fe_{4+x}S_3$, considering both the P-T effect on the variation of x and the effect of x on the thermoelastic properties of $Fe_{4+x}S_3$.

Constraints on the temperature for inner core crystallization can be obtained by examining the thermal stability of $Fe_{4+x}S_3$, which increases quite significantly with pressure, from less than 1200 K at 14 GPa[22] to ~1500 K at 21 GPa[17]. At higher temperatures, $Fe_{4+x}S_3$ will melt incongruently to form solid FeS and Fe-S liquid: $Fe_{4+x}S_3(solid) = 2FeS(solid) + Fe_{2+x}S(liquid)$[17]. The $Fe_{4+x}S_3$ synthesized in this study coexists with Fe-S liquid at 1640 K and 27 GPa, indicating a melting temperature higher than this. An extrapolation of the melting curve to the pressure at the center of the Martian core (~40 GPa), indicates that $Fe_{4+x}S_3$ is stable up to approximately 1970(±105) K (Supplementary Fig. 4). This relatively refractory behavior of $Fe_{4+x}S_3$ underlines the potential for it to form planetary inner cores.

The solidification regime of the Martian core will depend on the core's composition and temperature. We have parameterized the melting phase relations in the Fe-FeS system up to 40 GPa considering the liquidus phases Fe, $Fe_3S$, $Fe_{4+x}S_3$, and FeS and using our results and those from the literature (Supplementary Fig. 5). Since there is no experimental evidence to support the stability of $Fe_{12}S_7$ and $Fe_2S$[17,24] as liquidus phases under Martian core conditions, these phases were not considered in our model. Figure 4 shows Fe-S liquidus curves for different amounts of S, compared with Martian areotherms, from which the core solidification regime can be inferred by considering potential points of intersection. As illustrated in Fig. 5, during the cooling of Mars, if the core contains ~7–12 wt.% S, Fe-metal snow will form at the top of the Martian core, as proposed previously[18]. The crystalized Fe will sink but will be dissolved again in the deeper core region where the liquidus temperature is then lower than the areotherm (see curve

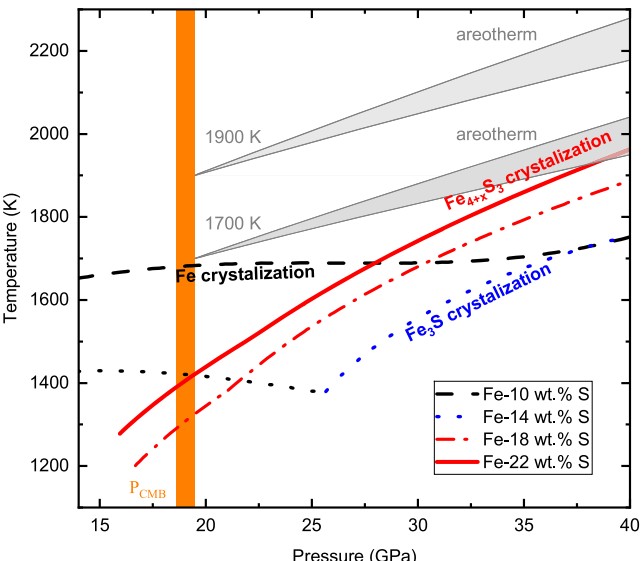

**Fig. 4 | Fe-S system liquidus curves as a function of pressure.** Black dashed line: Fe-10 wt.% S liquid where Fe is the liquidus phase; blue dotted line: Fe-14 wt.% S where $Fe_3S$ is the liquidus phase; black dotted line: Fe-14 wt.% S where Fe is the liquidus phase <25 GPa; red sold lines: Fe–18 wt.% S with $Fe_{4+x}S_3$ as the liquidus phase; red dash-dot line: Fe-22 wt.% S with $Fe_{4+x}S_3$ as the liquidus phase. The grey-shaded regions are areotherm estimates for the Martian core[16] with CMB temperatures at 1700 K and 1900 K.

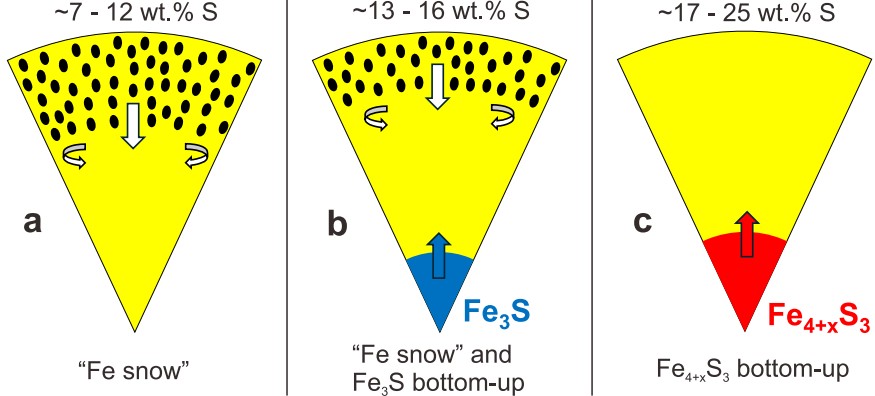

**Fig. 5 | Solidification regimes of the Martian core for different bulk core sulfur concentrations. a** Iron snow. **b** Simultaneous iron snow and $Fe_3S$ crystalizing from the center. **c** Bottom-up crystallization of $Fe_{4+x}S_3$.

10 wt.% S in Fig. 4). If the Martian core contains ~13–16 wt.% S, the liquidus curve will first decrease with pressure (see curve 14 wt.% S in Fig. 4), where Fe-metal is the liquidus phase. However, as the S content of the eutectic melt decreases with increasing pressure, $Fe_3S$ will replace Fe as the liquidus phase at higher pressures (21–32 GPa, depending on the exact S content), which will cause the liquidus temperature to increase with pressure. In this case, the areotherm could intersect the liquidus curve at both the top and bottom of the Martian core, which means that both iron snow and bottom-up growth of $Fe_3S$ could occur simultaneously. However, if the Martian core contains 17–25 wt.% S, which would be quite consistent with recent density estimates[1–3], $Fe_{4+x}S_3$ will be the liquidus phase over the entire pressure range of the Martian core, with a liquidus temperature that increases continuously with pressure (see curves 18 and 22 wt.% S in Fig. 4). This means that a $Fe_{4+x}S_3$ inner core will start to crystalize if the center of such a S-rich Martian core cools below approximately 1960 K.

Mars has no active global magnetic field, implying that there is no dynamo operating in the Martian core. Crystallization of Fe in the form of iron snow would cause chemical convection below the snow zone, although whether this convection would provide enough energy to run a dynamo is debated[15,16,35–37], as it is dependent on the generation of a positive net buoyancy flux[38]. In contrast, the crystallization of a sulfide inner core is consistent with the absence of an active dynamo, as the residual liquid would be richer in Fe and would, therefore, remain at the base of the outer core, inhibiting chemical convection[15].

The presence of an inner core composed of $Fe_{4+x}S_3$ would likely have a negligible impact on interpretations of geodetic observations. For instance, with a small inner core allowed by current seismic constraints (e.g., a radius of <600 km), the mass of a $Fe_{4+x}S_3$ inner core would constitute less than 5% of the total core mass, altering the moment of inertia by only ~0.1%. Additionally, according to numerical calculations, the nutation effects of an inner core with a radius of 600 km are expected to be insignificant[33]. Thus, geodesy alone may be insufficient to constrain the existence of a relatively small inner core, and further seismic observations from future space missions, as well as additional analyses of InSight seismic data, are needed to provide more definitive evidence regarding the presence or absence of a Martian inner core. Further experimental measurements to enable seismic velocities of $Fe_{4+x}S_3$ to be determined would be also important for the interpretation of potential inner core-related seismic signals.

The temperature at the center of the Martian core would need to be lower than 1800 K to allow $Fe_3S$ to crystallize or for an "iron snow" zone to reach the Martian center and form an inner core of Fe (Fig. 4). This corresponds to a temperature of approximately 1500–1600 K at the core-mantle boundary (CMB), which is lower than all current thermal models for Mars[34,39–43]. Therefore, crystallization of a $Fe_3S$ or Fe inner core are likely to be only future scenarios, possible only after Mars has cooled further[18]. On the other hand, the crystallization temperature of $Fe_{4+x}S_3$ in a core containing, for example, 22 wt.% S—an amount that could satisfy the Martian core's density deficit (e.g., the core density model from Irving et al.[3])—would be approximately 1960 (±105) K. This approaches the lower limit of the estimated temperature of the Martian core in some thermal models[39–41]. The detection of a Martian inner core through further geophysical observations, along with an estimate of its density, would provide critical constraints on the chemical composition and temperature of the Martian core. Moreover, the existence of a Martian inner core would imply a relatively cool Martian interior, which would be incompatible with the presence of a basal magma layer on top of the CMB. Conversely, if an inner core is confirmed to be absent, the $Fe_{4+x}S_3$ melting temperature, i.e., 1960 (±105) K, would provide a lower limit for the temperature at the center of Mars.

It is worth noting that the addition of other light elements, such as O, C, and H, may impact the crystallization phase relations of the Fe-FeS system, though the effects remain largely unknown due to a lack of experimental studies in more complex systems. On the one hand, the addition of multiple light elements could further lower the eutectic temperature of the system, making an inner core less likely. On the other hand, it is possible that elements such as H, that can be incorporated in high pressure sulfides in stoichiometric proportions[44], might partition sub-equally between liquid and melt phases and have little effect on crystallization temperatures. While this study provides a preliminary estimation of the likelihood of $Fe_{4+x}S_3$ crystallization in the Martian core, further experiments involving relevant more complex chemical compositions are needed to test the hypotheses proposed here.

## Methods
### Starting material
The initial mixtures were composed of metallic iron powder (99.9%, 10 μm particle size) and high-purity elemental sulfur (99.999%). The bulk compositions of the mixtures are listed in Supplementary Table 5. The elements were mixed in an agate mortar under ethanol for about 45 min, followed by drying overnight in an oven at 340 K. For the in-house multi-anvil experiments, the mixed powder was directly used as the starting material, without any pre-treatment. For the synchrotron multi-anvil experiments, the powder mixtures underwent a pre-sintering process. This involved compressing the powders in a piston-cylinder press at 0.5 GPa and 1000 K for over 6 h. Following sintering, the materials were machined into regular cylinders, each measuring approximately 1 mm in diameter and 0.6 mm in height. The sintered cylinders contained a mixture of metallic Fe and troilite (FeS). For the LH-DAC experiments, a pre-synthesized $Fe_{4+x}S_3$ crystal separated from a prior multi-anvil experiment was utilized. The degraded crystal was pre-compressed into a thin platelet, roughly 10 μm thick, to serve as the starting material.

### In-house multi-anvil experiments
10 mm and 7 mm edge length octahedral assemblies were compressed using tungsten carbide cubic anvils with 5 mm and 3 mm truncation edge lengths (TEL), respectively. The 10/5 assembly was used for experiments targeting pressures between 14 to 16 GPa. The 7/3 assembly was utilized for the experiments at 27 GPa[45,46]. In these in-house multi-anvil experiments, the pressure uncertainties at high temperatures are estimated to be ±2 GPa. The starting materials were loaded into a gold capsule for run S7995, while MgO capsules were used in all other runs. After reaching the target loads, the samples were heated to high temperatures using $LaCrO_3$ heaters. Temperatures were monitored using a type D W-Re thermocouple, except for run S7995 and I1691a, where the temperature was estimated based on the power-temperature relation from previous experiments. The pressure effects on the electromotive force (EMF) of the thermocouple were corrected[47]. These temperatures were maintained for durations ranging from 1 to 10 hours, followed by rapid quenching to room temperature by shutting down the power source.

### Synchrotron multi-anvil experiments
The experiments were conducted at the beamline PSICHE at the SOLEIL synchrotron[48]. This assembly was equipped with a boron-doped diamond (BDD) heater, synthesized through the chemical vapor deposition (CVD) method[49,50]. The CVD-BDD heaters are notable for providing stable heating conditions and high X-ray transparency, which assists the collection of high quality XRD data. The assemblies were compressed to high pressures using tungsten carbide anvils with 5 mm TEL. The samples, which were placed in a corundum ($Al_2O_3$) capsule, were illuminated by the white X-ray beams, and the energy-dispersive XRD patterns of the samples were collected in situ at high pressures and high temperatures. The unit cell parameters of the phases present in the samples were determined through Rietveld Le Bail fitting, using the GSAS-II software package[51]. Temperatures were

measured using a type D thermocouple, with corrections also applied for pressure effects on the EMF[47]. Pressures were evaluated using the equation of state of corundum[52]. The differences in pressure are less than 0.2 GPa when applying an alternative pressure standard for corundum[53].

### Single crystal syntheses in LH-DAC and high-pressure SC-XRD
We employed a BX90-type diamond anvil cell[54], equipped with a pair of diamond anvils each having a culet diameter of 250 µm for the high-pressure single crystal synthesis and measurements. Pre-indented rhenium gaskets were employed. A thin platelet of $Fe_{4+x}S_3$ sample was sandwiched between two KCl layers and compressed to approximately 15 GPa in run LJFeS01. KCl provided both thermal insulation and the pressure marker[55] in the experiment. The sample was then heated from both sides using near-infrared lasers to a temperature of 1150 (200) K, employing a modified version of the portable double-sided laser of the ID14 beamline at ESRF[56,57]. The temperature for LJFeS01 was estimated based on the melting phase relations in the high-pressure Fe-FeS system[22], as the $Fe_{4+x}S_3$ phase coexisted with Fe-S liquid (Supplementary Fig. 6). For run LD101, helium was loaded at 1.2 kbar and served both as a pressure-transmitting medium and thermal insulator for laser heating. The sample was compressed to ~20 GPa and then heated to 1300 (200) K, employing the in-house laser-heating system at the Bayerisches Geoinstitut[58], with temperatures estimated by fitting the radiation of the sample using a grey body approximation. The pressure of run LD101 was estimated based on a pressure calibration of the Raman shift of the diamond anvils[59]. For both runs, the samples were subjected to heating for durations ranging from 5 to 10 s before being rapidly quenched to room temperature.

High-pressure XRD measurements were conducted at the high-pressure diffraction beamline ID15B at the ESRF in Grenoble. We utilized a focused X-ray beam with a wavelength of 0.4100 Å and a beam size of approximately 1 µm × 1 µm. Initially, a 2D-scan XRD map with a step size of 2 µm was constructed by scanning the sample stage[60]. This process was aimed at locating phases of interest within the samples, as illustrated in Supplementary Fig. 6. Upon locating these phases, SC-XRD data collection was conducted over a range of −30° to +30° for LJFeS01, and −34° to +34° for LD101, with an increment step of 0.5°. The Domain Auto Finder program (DAFi)[61] was employed for the rapid identification of domains of $Fe_{4+x}S_3$ microcrystals within the complete SC-XRD dataset collected from the multiphase samples. Domains of $Fe_{4+x}S_3$ were primarily located in the region adjacent to the melt within the laser-heated spots. In the colder areas of the laser-heated spot, FeS III and several unidentified phases were indexed from the SC-XRD dataset. As this paper concentrates on the stability of liquidus phases, some further reflections for the unidentified subsolidus sulfide structures identified within the samples are outside of the scope of this discussion. Data reductions were performed using the CrysAlis Pro software package. The crystal structures of $Fe_{4+x}S_3$ from each run were subsequently solved and refined using the Olex2 software package[62]. Further details regarding the structure solution and refinement can be found in Supplementary Methods.

### Sample recovery and chemical analysis
After the completion of the in-house multi-anvil experiments, the run products were carefully recovered to ambient conditions, mounted in epoxy resin, and subsequently polished for chemical analyses. The samples were analyzed using a JEOL JXA-8200 electron probe microanalyzer (EPMA), which was operated at 15 kV and 15 nA. Calibration standards of metallic iron, pyrite, and periclase were employed for Fe, S, and O, respectively. The solid phases of the samples were analyzed with a focused beam, approximately 1 µm in diameter. The compositions of the quenched liquids were measured using a defocused beam with a diameter ranging from 10 to 30 µm, depending on the size of the quenched texture.

### Parameterization of liquidus temperature
The pressure and composition dependencies of the liquidus temperature are parameterized following a similar approach to previous models[16,63], incorporating more experimental constraints from this study. The melting phase diagram at a given pressure is constructed using the melting temperatures of the liquidus phases (Fe, $Fe_3S$, $Fe_{4+x}S_3$, and FeS) and the eutectic temperature and composition as anchor points. It is assumed that the liquidus temperature has a linear dependence on sulfur concentration when Fe or FeS is the liquidus phase and a parabolic dependence on sulfur concentration when $Fe_{4+x}S_3$ or $Fe_3S$ are the liquidus phase. Fe, $Fe_{4+x}S_3$, and FeS are liquidus phases at pressures from 14 to 21 GPa, while Fe, $Fe_3S$, $Fe_{4+x}S_3$, and FeS are liquidus phases at pressures above 21 GPa. We use the melting curves of Fe[64] and FeS[65] from the literature and evaluate the eutectic compositions and temperatures at high pressures (Supplementary Fig. 7), as well as the melting curves of $Fe_3S$ and $Fe_{4+x}S_3$ (Supplementary Fig. 4), using data from this study and the literature[17,22,66]. The change in the peritectic composition from $Fe_{4+x}S_3$ plus liquid to FeS plus liquid as a function of pressure was estimated based on data from this study and the literature[17,22]. The sulfur content is parameterized to increase with pressure until it reaches the composition of $Fe_5S_3$. Meanwhile, the peritectic composition at the point where $Fe_3S$ plus liquid reacts to $Fe_{4+x}S_3$ plus liquid is assumed to be the composition of $Fe_3S$ (16.1 wt.% S). The parameters describing the model are listed in Supplementary Table 6. The melting phase relations at 15 GPa, 21 GPa, 27 GPa, and 40 GPa, generated using this model, are plotted in Supplementary Fig. 5 and compared with literature data to demonstrate consistency.

## Data availability
Crystallographic data for the structures reported in this paper are available at the Cambridge Crystallographic Data Center under deposition numbers CCDC 2413544 and 2413545. Additional data supporting the main findings are provided in the supplementary materials. Raw data files can be obtained from the corresponding author upon request.

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

## Acknowledgements

The authors thank L. Yuan, R. Pierru, and E. Kubik for insightful discussions. We appreciate assistance from D. Krauße, A. Potzel, and D. Wiesner in chemical characterization using electron microscopy. R. Njul, A. Rother, H. Fischer, and S. Übelhack are acknowledged for their help with sample preparation and the maintenance of the large-volume presses at BGI. This research was supported by DFG grant FR1555/11 to D.J.F. We acknowledge the European Synchrotron Radiation Facility (ESRF) for provision of experiment time at the ID15B beamline and offline laser-heating facilities of the ID14 beamline. I.K. acknowledges funding provided by the European Union (ERC, LECOR, project number 101042572). Views and opinions expressed are, however, those of the authors only and do not necessarily reflect those of the European Union or the European Research Council. Neither the European Union nor the granting authority can be held responsible for them.

## Author contributions

L.M. conceived and designed this project. L.M., X.L., T.B.B., L.D., W.Z., J.C., A.N., I.K., G.A., A.K., M.H., N.G., L.H., and D.J.F. performed the experiments. W.Z., T.B.B., L.D., X.L., and L.M. analyzed the single-crystal X-ray diffraction data. L.M., D.J.F., X.L., A.N., O.N., I.K., and L.D. contributed to the interpretation of the results. L.M. and D.J.F. wrote the paper with contributions from all the authors.

## Funding

## Competing interests

The authors declare no competing interests.
