## [Peer review file · Nature Communications]

The structure and stability of $\text{Fe}_{4+x}\text{S}_3$ and its potential to form a Martian inner core

Corresponding Author: Mr Lianjie Man

Version 0:

Reviewer comments:

Reviewer #1

(Remarks to the Author)

Review of The structure and stability of $\text{Fe}_{4+x}\text{S}_3$ and its potential to form the inner core of Mars by Man et al. submitted to Nature Communications.

This paper studies properties of $\text{Fe}_{4+x}\text{S}_3$ phase which may be relevant to the martian inner core. They studied the structure of $\text{Fe}_{4+x}\text{S}_3$ using single crystal XRD techniques and found out that the previously discussed $\text{Fe}_{3+x}\text{S}_2$ are indeed $\text{Fe}_{4+x}\text{S}_3$. Separating the temperature and compositional effects on the unit-cell volume, they established P-V-T-x relations for the phase. They then discuss the $\text{Fe}_{4+x}\text{S}_3$ is denser than the martian core if it is the liquidus phase at the center of the planet, which does not kick-off convection of the liquid core, which can then explain the absence of magnetic fields. The motivation of the research is well articulated and the approach and strategy taken are reasonably good. I think the major outcome is that the absence of magnetic fields does not necessarily indicate the absence of a solid inner core. I support publication of this manuscript in Nature Communications. My comments and suggestions are appended below.

1. Single crystal XRD

How many grains were picked by X-rays? Just one grain? If it is one grain, is there any chance of missing some reflections due to the limitation of the scanned range, i.e., a range of -30 degrees to 30 degrees for LJFeS_3 , and -34 degrees to 34 degrees for LD101 ?

2. T dependence of the volume and x of $\text{Fe}_{4+x}\text{S}_3$

This is one of the highlights of this paper. The authors claim that $\text{Fe}_{4+x}\text{S}_3$ showed an increase in x value and expands the volume when the temperature was increased. I'm not sure how incorporating Fe expands the unit-cell volume. What is the bonding force between Fe and S at this composition? I like to see how the average atomic volume responds to changes in x.

Minors list

Line 178-179: The crystal structure of Fe_3S_2 phase was resolved to be orthorhombic in Zhao et al. (2024).

Figure 3b: I assume that x for this study and for Zhao et al. are different. Can you confirm?

Figure 4 caption: Helffrich

Extended Data Fig. 4: Why not drawing subsolidus phase relations, with the composition of $\text{Fe}_{4+x}\text{S}_3$ being temperature dependent.

Extended Data Fig. 4: You should mention that you simplified the Fe-liquidus curves which are supposed to be sigmoidal shaped, although this may not have significant impacts on the discussions.

L212, $(x=9.5 \times 10^{-4} \times (T-750))$: Give the uncertainty in the fitting.

On this fitting, L238 – 239 (The red dashed line represents the linear fit of the x-T relationship when coexisting with metallic iron):

Which data did you fit? Obviously Fei et al.'s data are not on the red line.

Line 219, ($5.3 \pm 2.0 \times 10^{-5} \text{ K}^{-1}$): Is this for 1 bar?

L225-228, the density of $\text{Fe}_{4+x}\text{S}_3$

This is a very important discovery of this paper. Can you define the EoS you used? The second-order Birch-Murnaghan EoS was used. But there are no descriptions of the EoS. How did you incorporate the thermal effects in the EoS, high-T B-M or the Mie-Grueneisen thermal pressure model?

L245: Is 940K meant to be 950 K (See Figure 3b)?

L248, (The temperature stability field of $\text{Fe}_{4+x}\text{S}_3$):

Define the chemical reaction for the high-temperature stability of $\text{Fe}_{4+x}\text{S}_3$.

L320-321: Give the proportions of Fe and S for the starting mix.

Reviewer #2

(Remarks to the Author)

This manuscript introduces a newly synthesized phase, $\text{Fe}_{4+x}\text{S}_3$, formed under conditions simulating the Martian core. The paper characterizes the crystal structure of this new phase and suggests that the previously identified phases, Fe_3S_2 and $\text{Fe}_{3+x}\text{S}_2$, share a similar structure. Notably, $\text{Fe}_{4+x}\text{S}_3$ exhibits a higher density than the liquid Martian core, and its liquidus may intersect with the areotherm. This finding supports the potential existence of a Martian inner core, assuming that the Martian core contains 17-25 wt. % sulfur, leading to crystallization at the center. In general, I think this is a very interesting and important experiment result and has great implications for the Martian core evolution. Thus, I would think this work is suitable for Nature Communications.

As I am not an expert in mineral physics experiments, I will focus my comments on the broader implications of the new findings. There is no doubt that the identification of this new phase is significant. However, given the current uncertainty regarding the existence of a Martian inner core, it is challenging to fully assess the implications on the formation of the Martian inner core. A more in-depth discussion of several key issues would enhance the impact of the findings.

1. The existence of an inner core would require a central temperature of less than 1960 (± 105) K, suggesting that the temperature at the CMB is ~ 1700 K. If this is the case, it would be important to comment on the formation of a basal silicate magma layer, which likely requires a higher CMB temperature. In other words, if there is a basal silicate magma layer, are we able to have an inner core even with the new $\text{Fe}_{4+x}\text{S}_3$?
2. The proposed formation of a bottom-up inner core also implies a significant sulfur content, which appears to contradict geochemical/cosmochemical constraints indicating lower sulfur values. While a sulfur content of 17-25 wt.% is supported by seismic constraints for a low-density liquid core, most studies also pointed out the significance of other light elements, such as O, C, and H. An in-depth discussion on the contributions of these elements and how to reconcile the apparent contradiction between this result and geochemical data would be beneficial.
3. If the new $\text{Fe}_{4+x}\text{S}_3$ constitutes the inner core with a density of 7.5 g/cm^3 , this density is $>10\%$ greater than the highest current estimates for liquid core density. It would be intriguing to explore how this density contrast might affect calculations of mass, moment of inertia, and k_2 for inner core radii of 100, 300, and 600 km... Such an analysis could provide insights into whether current geodetic measurements can definitely rule out the presence of an inner core. Additionally, a brief discussion on the seismic velocity properties associated with this new phase would enhance the manuscript's relevance.
4. Alternatively, You might begin by outlining the current understanding of the Martian core, subsequently incorporating the new phase to emphasize its thermal profile and composition. From there, expand the discussion to explore the potential for a solid inner core.

Some other minor comments:

1. Line 76-78: I think the correct reference here should be Fei et al. (2000).
2. Line 216-217: Here, you assume that the thermal expansion coefficient does not vary significantly with x . How good is this assumption?
3. Line 221: If I treat Fe_3S_2 exactly the same as $\text{Fe}_{4+x}\text{S}_3$, why do their thermal expansions differ so significantly? A discussion on this point would be helpful.
4. Line 284-286: I find this statement to be somewhat inaccurate. While the chemical convection introduced by the crystallization of an inner core is important for the development of a dynamo. However, it does not guarantee a dynamo. A positive net buoyancy flux, accounting for thermal, composition, and other effects, is necessary to allow a dynamo. Thus, the absence of an active dynamo may not be against the possibility of one existing; it merely indicates that conditions are not currently favorable.

You may refer Hemingway & Driscoll (2021) and the references cited therein for more details.

Hemingway, D. J., & Driscoll, P. E. (2021). History and future of the Martian dynamo and implications of a hypothetical solid inner core. *Journal of Geophysical Research: Planets*, 126, e2020JE006663. <https://doi.org/10.1029/2020JE006663>

Version 1:

Reviewer comments:

Reviewer #1

(Remarks to the Author)

This is a revised manuscript. The authors reasonably addressed my concerns and the manuscript has been much improved. I'm happy to support publication of this version in Nature Communications. This will be a timely and high-impact publication.

Reviewer #2

(Remarks to the Author)

The authors have adequately addressed my comments, and I believe the manuscript is now suitable for acceptance in its current form.

Reviewer #1 (Remarks to the Author):

Review of The structure and stability of Fe_{4+x}S₃ and its potential to form the inner core of Mars by Man et al. submitted to Nature Communications.

This papers studies properties of Fe_{4+x}S₃ phase which may be relevant to the martian inner core. They studied the structure of Fe_{4+x}S₃ using single crystal XRD techniques and found out that the previously discussed Fe_{3+x}S₂ are indeed Fe_{4+x}S₃. Separating the temperature and compositional effects on the unit-cell volume, they established P-V-T-x relations for the phase. They then discuss the Fe_{4+x}S₃ is denser than the martian core if it is the liquidus phase at the center of the planet, which does not kick-off convection of the liquid core, which can then explain the absence of magnetic fields. The motivation of the research is well articulated and the approach and strategy taken are reasonably good. I think the major outcome is that the absence of magnetic fields does not necessarily indicate the absence of a solid inner core. I support publication of this manuscript in Nature Communications. My comments and suggestions are appended below.

1. Single crystal XRD

How many grains were picked by X-rays? Just one grain? If it is one grain, is there any chance of missing some reflections due to the limitation of the scanned range, i.e., a range of -30 degrees to 30 degrees for LJFeS01, and -34 degrees to 34 degrees for LD101?

It is indeed inevitable that some reflections are not captured due to the limited opening angle in all diamond anvil cell experiments and due to the grain orientations. As shown in Supplementary Table 2., the coverage of independent reflections is 39.4% for the grain analyzed in LJFeS01 and 58.1% for the grain in LD101. However, more than 10 different grains of Fe_{4+x}S₃ crystals were indexed using the software DAFi (Aslandukov et al., 2022, doi.org/10.1107/S1600576722008081) in each run. The structural data obtained from several grains with different orientations (as specified at lines 133-139) show strong consistency with each other, giving us full confidence in the structural solution and refinement. For clarity, we have only presented examples with the highest data quality in the manuscript. Note that the single-crystal data, in spite of the limited data coverage, constrain much better than powder data the space group of this phase, since the reflection conditions can be verified for several reflections given the three-dimensional coverage versus one-dimensional for powder data. For example, it is clearly visible in the deposited CIFs that all *0kl* reflections observed with *k+l = odd* have negligible intensities, whereas those with *k+l = even* have large intensities indicating clearly that the space group is *Pnma*. This information is now better reported at lines 141-142.

2. T dependence of the volume and x of Fe_{4+x}S₃

This is one of the highlights of this paper. The authors claim that Fe_{4+x}S₃ showed an increase in x value and expands the volume when the temperature was increased. I'm not sure how incorporating Fe expands the unit-cell volume. What is the bonding force between Fe and S at this composition? I like to see how the average atomic volume responds to changes in x.

We have now added an further section of text in the Supplementary Discussion which explains

how the increase of the occupancy of Fe into the tetrahedral site increases the tetrahedral volume significantly and as a consequence also affects the other polyhedral sites. This requires some consideration of the compressibility because the single-crystal data have been collected at different pressure. Nevertheless, the effect is clearly visible in Supplementary Fig. 1 where the unit-cell volume of the Fe-rich sample is compared with the compression curve of the Fe-poor sample at room temperature. We have also changed the main text at lines 170-174 to explain this effect more clearly

Minors list

Line 178-179: The crystal structure of Fe₃S₂ phase was resolved to be orthorhombic in Zhao et al. (2024).

We added this information in lines 105 to 106 of the manuscript.

Figure 3b: I assume that x for this study and for Zhao et al. are different. Can you confirm?

The x value increases with temperature, as shown in the x-T relationship plotted in Fig. 3a. The red dashed line in Fig. 3a represents the x-T relationship when Fe_{4+x}S₃ coexists with metallic Fe at approximately 16 GPa. Since the experiments by Zhao et al. (2024) were conducted at similar pressures (~13 to 16 GPa), we assumed that Zhao et al.'s data follow the same x-T relationship. Additionally, the P-V-T data of Fe_{4+x}S₃ reported by Zhao et al. (2024) align well with our *in situ* HP-HT data at around 14 GPa (Fig. 3b), further supporting that Zhao et al.'s data follow the same x-T relationship as observed in this study. We have clarified this in the caption of Fig 3. (lines 1035 to 1036 of the manuscript).

Figure 4 caption: Helffrich

Changed.

Extended Data Fig. 4: Why not drawing subsolidus phase relations, with the composition of Fe(4+x)S₃ being temperature dependent.

In the present study, our data on subsolidus phase relations are limited to pressures around 16 GPa. As noted in lines 244 to 247, there may be a pressure effect on the variation of x, though we cannot quantify this effect based on the current dataset. This figure is focused on illustrating the melting phase relations of Fe-FeS. To maintain consistency across the sub-figures and to avoid potential misinterpretations, we chose not to include the details of the subsolidus phase relations.

Extended Data Fig. 4: You should mention that you simplified the Fe-liquidus curves which are supposed to be sigmoidal shaped, although this may not have significant impacts on the discussions.

This is added in the caption of Supplementary Fig. 5.

L212, ($x=9.5 \times 10^{-4} \times (T-750)$): Give the uncertainty in the fitting.

Added

On this fitting, L238 – 239 (The red dashed line represents the linear fit of the x - T relationship when coexisting with metallic iron):

Which data did you fit? Obviously Fei et al.'s data are not on the red line.

We fitted the volume data of $\text{Fe}_{4+x}\text{S}_3$ coexisting with metallic Fe at pressures between 15 and 16 GPa, using data from this study and the literature (Urakawa et al., 2018; Tsuno et al., 2009). We clarified this in lines 253 of the manuscript.

Line 219, ($5.3 \pm 2.0 \times 10^{-5} \text{ K}^{-1}$): Is this for 1 bar?

This is the thermal expansion coefficient for the available data at pressures from 13 to 16 GPa and temperatures from 800 to 1100 K. The thermal expansion coefficient was assumed to be constant over this limited pressure and temperature range. This is now explained in lines 263 to 264 of the manuscript.

L225-228, the density of $\text{Fe}_{4+x}\text{S}_3$

This is a very important discovery of this paper. Can you define the EoS you used? The second-order Birch-Murnaghan EoS was used. But there are no descriptions of the EoS. How did you incorporate the thermal effects in the EoS, high- T B-M or the Mie-Grueneisen thermal pressure model?

The cold compression behavior was described using a second-order Birch-Murnaghan EoS. However, the volume of $\text{Fe}_{4+x}\text{S}_3$ at high pressure and high temperature (HP-HT) can only be accurately modeled with a P-V-T- x EoS, where x varies with temperature (and potentially pressure). Due to insufficient data, we cannot fully constrain how thermal expansion changes with pressure. Therefore, as noted in our previous response, we treated the thermal expansion coefficient as a constant. As explained in lines 322 to 325, this may lead to a slight underestimation of $\text{Fe}_{4+x}\text{S}_3$'s density when extrapolating the EoS to the center of the Martian core; however, this does not alter our conclusion that $\text{Fe}_{4+x}\text{S}_3$ would be gravitationally stable at Mars' core center. Extensive in situ HP-HT experiments will be needed in future studies to establish a comprehensive thermodynamic model for $\text{Fe}_{4+x}\text{S}_3$ and to provide a detailed P-V-T- x EoS.

L245: Is 940K meant to be 950 K (See Figure 3b)?

No. The 940 K (red dashed line) is the value reported in Zhao et al. (2024).

L248, (The temperature stability field of $\text{Fe}_{4+x}\text{S}_3$):

Define the chemical reaction for the high-temperature stability of $\text{Fe}_{4+x}\text{S}_3$.

The chemical reaction is added (lines 332 to 333 of the manuscript).

L320-321: Give the proportions of Fe and S for the starting mix.

The bulk compositions of the starting materials are listed in Supplementary Table 5., and now we add a note in the methods (lines 435 to 436 of the manuscript).

Reviewer #2 (Remarks to the Author):

This manuscript introduces a newly synthesized phase, $Fe_{4+x}S_3$, formed under conditions simulating the Martian core. The paper characterizes the crystal structure of this new phase and suggests that the previously identified phases, Fe_3S_2 and $Fe_{3+x}S_2$, share a similar structure. Notably, $Fe_{4+x}S_3$ exhibits a higher density than the liquid Martian core, and its liquidus may intersect with the areotherm. This finding supports the potential existence of a Martian inner core, assuming that the Martian core contains 17-25 wt. % sulfur, leading to crystallization at the center. In general, I think this is a very interesting and important experiment result and has great implications for the Martian core evolution. Thus, I would think this work is suitable for Nature Communications.

As I am not an expert in mineral physics experiments, I will focus my comments on the broader implications of the new findings. There is no doubt that the identification of this new phase is significant. However, given the current uncertainty regarding the existence of a Martian inner core, it is challenging to fully assess the implications on the formation of the Martian inner core. A more in-depth discussion of several key issues would enhance the impact of the findings.

1. The existence of an inner core would require a central temperature of less than 1960 (± 105) K, suggesting that the temperature at the CMB is ~ 1700 K. If this is the case, it would be important to comment on the formation of a basal silicate magma layer, which likely requires a higher CMB temperature. In other words, if there is a basal silicate magma layer, are we able to have an inner core even with the new $Fe_{4+x}S_3$?

The conditions that would allow for a Martian inner core are incompatible with the presence of a basal magma layer, and vice versa. We have added this clarification in the Discussion section of the manuscript (lines 403 to 405 of the manuscript).

2. The proposed formation of a bottom-up inner core also implies a significant sulfur content, which appears to contradict geochemical/cosmochemical constraints indicating lower sulfur values. While a sulfur content of 17-25 wt.% is supported by seismic constraints for a low-density liquid core, most studies also pointed out the significance of other light elements, such as O, C, and H. An in-depth discussion on the contributions of these elements and how to reconcile the apparent contradiction between this result and geochemical data would be beneficial.

We propose the possibility that sulfur may vary independently of elements with similar condensation temperatures, which could help reconcile the apparent contradiction between the high sulfur content suggested by our results and the predictions of geochemical models (lines 62 to 68 of the manuscript). Furthermore, we have expanded the discussion to include perspectives on the potential contributions of other light elements, such as O, C, and H, and their role in Martian core composition (lines 413 to 422 of the manuscript).

3. If the new $Fe_{4+x}S_3$ constitutes the inner core with a density of 7.5 g/cm^3 , this density is $>10\%$ greater than the highest current estimates for liquid core density. It would be intriguing to explore how this density contrast might affect calculations of mass, moment of inertia, and k_2 for inner core radii of 100, 300, and 600 km... Such an analysis could provide insights into whether current

geodetic measurements can definitely rule out the presence of an inner core. Additionally, a brief discussion on the seismic velocity properties associated with this new phase would enhance the manuscript's relevance.

Overall, we estimate that the effect of a potential $\text{Fe}_{4+x}\text{S}_3$ inner core on geodetic measurements would be minimal. This is primarily because current seismic evidence constrains the potential size of a Martian inner core to be relatively small (with a radius of <750 km). Therefore, further constraints from seismic data are the most promising approach to confirm its existence. We have expanded the discussion regarding this aspect in lines 377 to 387 of the manuscript.

Modeling k_2 is not straightforward, as it requires additional information such as the viscoelastic properties of each core layer, which remain largely unknown. Addressing this is beyond our current expertise. Additionally, we could not provide P-wave and S-wave velocities for $\text{Fe}_{4+x}\text{S}_3$ from this study or the existing literature, as future measurements, such as ultrasonic interferometry or inelastic X-ray scattering, would be necessary. For these reasons, implications regarding k_2 and seismic velocities are not discussed in the manuscript.

4. Alternatively, You might begin by outlining the current understanding of the Martian core, subsequently incorporating the new phase to emphasize its thermal profile and composition. From there, expand the discussion to explore the potential for a solid inner core.

Thank you for the insightful suggestions, we restructured the discussion of the manuscript as suggested.

Some other minor comments:

1. Line 76-78: I think the correct reference here should be Fei et al. (2000).

Changed.

2. Line 216-217: Here, you assume that the thermal expansion coefficient does not vary significantly with x. How good is this assumption?

The influence of x on the thermal expansion coefficient of $\text{Fe}_{4+x}\text{S}_3$ cannot be accessed based on the data points in this study or from literature. As the observed volume changes with temperature are caused by both variations in x and thermal expansion, a very large amount of in situ HP-HT experiments would be required to fully separate these effects and constrain a full P-V-T-x equation of state. This is far beyond the scope of the current study. Sentences emphasizing the limitations of the current work are added at lines 325 to 327 of the manuscript. We have also emphasized that the assumption of a constant thermal expansion can only really lead to underestimates of the inner core density so this does not affect our conclusions- lines 322-325.

3. Line 221: If I treat Fe_3S_2 exactly the same as $\text{Fe}_{4+x}\text{S}_3$, why do their thermal expansions differ so significantly? A discussion on this point would be helpful.

The reason Zhao et al. reported an unusually large thermal expansion coefficient is that they assumed a constant stoichiometry, simplifying the composition to “ Fe_3S_2 ” ($\text{Fe}_{4.5}\text{S}_3$) without accounting for changes in x. In $\text{Fe}_{4+x}\text{S}_3$, both compositional variation (changing x) and thermal expansion contribute to volume expansion. Therefore, assuming all volume expansion is due

to thermal effects alone results in a substantial overestimation of the thermal expansion coefficient. We clarified this point in lines 268 to 270 of the manuscript.

4. Line 284-286: I find this statement to be somewhat inaccurate. While the chemical convection introduced by the crystallization of an inner core is important for the development of a dynamo. However, it does not guarantee a dynamo. A positive net buoyancy flux, accounting for thermal, composition, and other effects, is necessary to allow a dynamo. Thus, the absence of an active dynamo may not be against the possibility of one existing; it merely indicates that conditions are not currently favorable.

You may refer Hemingway & Driscoll (2021) and the references cited therein for more details.

*Hemingway, D. J., & Driscoll, P. E. (2021). History and future of the Martian dynamo and implications of a hypothetical solid inner core. *Journal of Geophysical Research: Planets*, 126, e2020JE006663. <https://doi.org/10.1029/2020JE006663>*

We did not claim that any crystallization regime of the Martian core would guarantee a dynamo, but only stated the possibility in the original manuscript. To clarify our points, we added more explanation as suggested by the reviewer at lines 372 to 373 in the manuscript.